# Retrovirus-Derived RTL/SIRH: Their Diverse Roles in the Current Eutherian Developmental System and Contribution to Eutherian Evolution

**DOI:** 10.3390/biom13101436

**Published:** 2023-09-22

**Authors:** Tomoko Kaneko-Ishino, Fumitoshi Ishino

**Affiliations:** 1Faculty of Nursing, School of Medicine, Tokai University, Kanagawa 259-1193, Japan; 2Center for Experimental Animals, Institute of Research, Tokyo Medical and Dental University (TMDU), Tokyo 113-8510, Japan

**Keywords:** *PEG10*, *PEG11/RTL1*, RTL/SIRH genes, placenta, brain, innate immunity, human disease, mammalian development and evolution

## Abstract

Eutherians have 11 retrotransposon Gag-like (RTL)/sushi-ichi retrotransposon homolog (SIRH) genes presumably derived from a certain retrovirus. Accumulating evidence indicates that the RTL/SIRH genes play a variety of roles in the current mammalian developmental system, such as in the placenta, brain, and innate immune system, in a eutherian-specific manner. It has been shown that the functional role of *Paternally Expressed 10* (*PEG10*) in placental formation is unique to the therian mammals, as are the eutherian-specific roles of *PEG10* and *PEG11/RTL1* in maintaining the fetal capillary network and the endocrine regulation of *RTL7/SIRH7* (aka *Leucine Zipper Down-Regulated in Cancer 1* (*LDOCK1*)) in the placenta. In the brain, *PEG11/RTL1* is expressed in the corticospinal tract and hippocampal commissure, mammalian-specific structures, and in the corpus callosum, a eutherian-specific structure. Unexpectedly, at least three RTL/SIRH genes, *RTL5/SIRH8*, *RTL6/SIRH3*, and *RTL9/SIRH10*, play important roles in combating a variety of pathogens, namely viruses, bacteria, and fungi, respectively, suggesting that the innate immunity system of the brain in eutherians has been enhanced by the emergence of these new components. In this review, we will summarize the function of 10 out of the 11 RTL/SIRH genes and discuss their roles in eutherian development and evolution.

## 1. Introduction

*Paternally expressed 10* (*PEG10*) [1] and *PEG11/Retrotransposon Gag-like 1* (*RTL1*) [2], together with *syncytin* [3,4], opened the door to a new field of research on retrovirus-derived genes in mammalian development and evolution. The fact that these are all essential endogenous genes despite their retroviral origin [3,4,5,6,7,8] has had a huge impact not only on genome biology but also on developmental and evolutionary biology, because retrotransposons, including endogenous retroviruses (ERVs), had long been construed to be “junk” in the mammalian genome. 

The discovery of retroviral *envelop* (*ENV*)-derived *syncytin* in primates stimulated the search for similar genes in primates as well as other lineages of eutherians and marsupials [9,10,11,12,13,14,15,16,17,18]. The discovery of essential functions for the retroviral *GAG* and *POL*-derived *PEG10* and *PEG11/RTL1* [5,6,19] led to the further screening of RTL/sushi-ichi retrotransposon homolog (SIRH) genes in eutherians (Figure 1) [5,20,21,22].

To date, 10 out of the 11 RTL/SIRH genes, including *PEG10* and *PEG11/RTL1,* have been shown to play important roles in eutherian development, such as in the placenta, brain, and innate immune system (Table 1) [26,27,28,29,30,31]. In addition, important roles for *PEG11/RTL1* in muscle development [32] and brain function [33], as well as an important role for *PEG10* in the placenta in mid to late gestation [34] have been elucidated. *PEG10* has also been implicated as the causative gene in Angelman syndrome [35] and amyotrophic lateral sclerosis (ALS) [36,37]. In this review, we focus on the RTL/SIRH genes, summarizing recent advances in their investigation, and discuss their roles in eutherian development and evolution.

## 2. Discovery of *PEG10* and *PEG11/RTL1* from Genomic Imprinting Research

Genomic imprinting, a term which describes the functional differences between the paternally and maternally derived genomes, was discovered by pronuclear transplantation experiments in mice in 1984. Three groups independently demonstrated that parthenogenetic embryos with two maternally derived genomes exhibit early embryonic lethality due to severe placental defects, while androgenetic embryos with two paternally derived genomes exhibit severe embryonic growth retardation associated with an overgrown placenta [38,39,40]. Extensive genetic analysis of mice with Robertsonian translocations of specific chromosomal loci also revealed functional differences between the parental chromosomes associated with early to late embryonic and postanal lethality as well as growth abnormalities [41,42]. The presence of imprinted genes with monoallelic paternal or maternal expression causes these genomic imprinting phenotypes [43,44,45,46,47].

*PEG10* and *PEG11/RTL1* have been identified as paternally expressed imprinted genes on human chromosome 7q21 and its orthologous mouse proximal chromosome 6 [1,48], and on the distal end of sheep chromosome 18 [2], respectively. It is known that maternal duplication of proximal chromosome 6 results in early embryonic lethality [42]. *Peg10* is responsible for this early embryonic lethal phenotype as well as parthenogenetic death due to severe placental dysplasia [5]. An inheritable form of muscular hypertrophy, called the callipyge phenotype, is mapped to the distal end of sheep chromosome 18 [49]. In humans and mice, paternal and maternal duplication of its orthologous imprinted region, human chromosome 14 and mouse distal chromosome 12, cause Kagami–Ogata (KOS14) and Temple syndromes (TS14), two genomic imprinting disorders, and late embryonic/neonatal lethal phenotypes associated with growth abnormalities, respectively [42,50,51,52,53,54]. *PEG11/RTL1*, together with *Delta-like 1 homologue* (*DLK1*), are the major genes responsible for KOS14 and TS14 as well as the abnormal phenotypes in mice caused by paternal and maternal duplication of distal chromosome 12 [6,18,32,33,55,56,57]. In sheep, *PEG11/RTL1* and *DLK1* also cause the callipyge phenotype [58,59]. Thus, *PEG10* and *PEG11/RTL1* are the genes responsible for certain abnormal imprinting phenotypes in eutherian mammals.

Both *PEG10* and *PEG11/RTL1* have homology to GAG and POL of the sushi-ichi long terminal repeat (LTR) retrotransposon [1,2,23,24,60]. Therefore, they were originally thought to be derived from the sushi-ichi retrotransposon, and, thus, were named RTL and/or SIRH. However, it is reasonable to assume that they were originally derived from the GAG and POL of a certain extinct retrovirus having a high degree of homology to the suchi-ichi retrotransposon [29,61], since *PEG10* arose in a common therian ancestor and *PEG11/RTL1* and the other RTL/SIRH genes also arose in a common eutherian ancestor [22,62,63], while the gypsy retrotransposon, which includes the sushi-ichi retrotransposon, is an infectious retrovirus in *Drosophila melanogaster* [64,65].

### 2.1. Roles of PEG10 and PEG11/RTL1 in Placental Evolution in Mammals 

As we have already reviewed the essential roles of *PEG10* and *PEG11/RTL1* in the placenta elsewhere [22,43,47,61,66,67], we here briefly summarize these points and focus on another placental role of *PEG10* as well as the possible interaction with *PEG11/RTL1*, and also discuss their roles in the evolution of the placenta in mammals.

The PEG10 protein is expressed in all of the trophoblast cell lines in the placenta. Paternal transmission of the *Peg10* KO allele (hereafter referred to as *Peg10* KO) causes early embryonic lethality due to poor placental growth associated with a complete lack of the labyrinth and spongiotrophoblast layers, because only the paternal allele of *Peg10* is active, while its maternal allele is repressed by the genomic imprinting mechanism [5]. As the labyrinth layer is an essential part of the placenta, where nutrient and gas exchange occur between fetal and maternal blood cells, *Peg10* KO embryos cannot grow beyond 9.5 days post coitus (dpc).

Paternal transmission of the *Peg11/RTL1* KO allele (hereafter referred to as *Peg11/Rtl1* Pat-KO) causes late fetal/neonatal lethality associated with late fetal growth retardation, while maternal transmission of the *Peg11/RTL1* KO allele (hereafter referred to as *Peg11/Rtl1* Mat-KO) causes neonatal lethality associated with abnormal fetal growth due to the overexpression of *Peg11/Rtl1* [6,55] This is because of the presence of maternally expressed *antiPeg11/antiRtl1,* a non-coding RNA encoding 7 microRNAs (miRNAs) that target *Peg11/Rtl1* mRNA via an RNAi mechanism [68,69,70]. The PEG11/RTL1 protein is restricted to expression in the endothelial cells of the fetal capillaries in the labyrinth layer of the placenta. Severe abnormalities of the fetal capillaries were observed in both the *Peg11/Rtl1* Pat- and Mat-KO placenta. In the former case, fetal capillary endothelial cells were clogged at many sites by an attack from surrounding trophoblast cells, while in the latter, the surrounding trophoblast cells were severely damaged, indicating that the PEG11/RTL1 protein plays an essential role in maintaining the feto–maternal interface of the placenta during pregnancy.

PEG10 retains the CCHC RNA-biding motif in GAG-like ORF1 and the DSG viral aspartic protease motif in POL-like ORF2. Unexpectedly, in contrast to the *Peg10* KO mice, the mice with the mutated DSG motif exhibit perinatal lethality (Figure 2) [34], as with the *Peg11/Rtl1* Pat- and Mat-KO mice described above. A point mutation introduced into the PEG10 DSG motif by replacing aspartic acid (D) with an alanine (A) residue using the CRISPR-Cas9 system (hereafter referred to as *Peg10*-ASG mutant) resulted in the loss of the self-cleavage activity of PEG10. These *Peg10*-ASG mutant mice exhibited embryonic and placental growth retardation from around 12.5 dpc, and about half of them died at 18.5 dpc (Figure 2A). Severe inflammation was detected around the fetal vasculature in the labyrinth layer of the mutant placenta (Figure 2B) [34]. In the labyrinth layer, PEG10 is expressed in all three layers of trophoblast cells, the two layers of syncytiotrophoblast (SynT-I and II) cells, and one layer of mononucleated sinusoidal trophoblast giant cells (s-TGCs) that surround the fetal capillary endothelial cells where PEG11/RTL1 is expressed [6,55]. 

These results demonstrate that not only the PEG11/RTL1 expression in endothelial cells but also the PEG10 expression in the surrounding trophoblast cells are essential for the maintenance of the fetal capillaries during mid to late gestation, although exactly what PEG10 and PEG11/RTL1 are doing there remains elusive at present. It is also apparent that *PEG10* has multiple essential functions in the placenta, and two of these functions, placental formation and maintenance of its fetal capillaries, were critical for the emergence of the chorioallantoic placenta in eutherian mammals. *PEG10* is a therian-specific gene [61] and, therefore, corresponds to the emergence of viviparity in therian mammals: Marsupials have a choriovitelline (yolk sac) placenta [71], while eutherians have evolved a chorioallantoic placenta that allows for longer gestation via the generation of a novel feto–maternal interface, presumably through a collaborative interaction between PEG10 and the newly domesticated (exapted) eutherian-specific PEG11/RTL1. 

### 2.2. Roles of PEG11/RTL1 in Muscle Development 

*PEG11/RTL1* is one of the major causative genes for the imprinting diseases KOS14 and TS14, which are caused by abnormal regulation of the imprinting region, that is, paternal and maternal disomy of human chromosome 14, respectively [19,50,51,52,53,54]. The former is characterized by neonatal lethality with respiratory failure, placentomegaly, polyhydramnios, developmental delay and/or intellectual disability, as well as feeding difficulties [51,53], whereas the latter is characterized by prenatal and postnatal growth retardation, feeding difficulties, muscle hypotonia, motor delay, early onset of puberty, and mild intellectual disability [52,54]. Their phenotypes are quite different, but their sites of damage are quite similar, i.e., in the muscle and brain as well as the placenta.

Importantly, *Peg11/Rtl1* Mat- and Pat-KO mice that, respectively, overexpress and lack PEG11/RTL1 expression, are very good models for KOS14 and TS14, not only in the placenta [6,19] but also in the muscle and brain as well (see next section) [32,33]. In skeletal muscle, expression of the PEG11/RTL1 protein is restricted to the late fetal and neonatal stages, and is, therefore, not detectable after 2 weeks of age, even in adults [32]. In the neonatal stage, *Peg11/Rtl1* Mat-KO mice had a significantly larger muscle fiber size, while *Peg11/Rtl1* Pat-KO mice had significantly thinner muscle fibers. However, after fixation, the muscle fibers of the *Peg11/Rtl1* Mat-KO mice exhibited severe shrinkage and detachment from the extracellular matrix (ECM) muscle, indicating that it is immature and more fragile than normal muscle (Figure 3A). In vivo experiments with cultured cells indicated that PEG11/RTL1 affects the proliferation of satellite cells (SCs) and the structural strength of SC-differentiated myoblasts, because myoblasts differentiated from SCs of Pat- and Mat-KO mice clearly exhibited weak or low myoblast structural strength [32]. This is consistent with the muscle-related defects in the KOS 14 and TS14 patients, such as respiratory failure and feeding difficulties in the former [51,53] and feeding difficulties, muscle hypotonia, and motor delay in the latter [52,54].

In myocytes, PEG11/RTL1 partially colocalizes with DESMIN, a component of the sarcomere cytoskeleton that connects the sarcomere to membranes of the sarcolemma and the nucleus at the Z-disc (Figure 3B) [32], thus, acting as the force-generating machinery in muscle. This suggests that PEG11/RTL1 plays some role in stabilizing the muscle contractile apparatus and/or regulating muscle contraction in the fetal/neonatal muscle fibers.

We assume that the intrinsically weak muscular strength and reduced movement of the fetus is necessary for a safe pregnancy because it is beneficial to both the mother and fetus. Then, PEG11/RTL1 may be required for such regulation by interacting or interfering with DESMIN and is, therefore, a well conserved feature of eutherians, suggesting that PEG11/RTL1 expression in the fetal muscle is an adaptation to the long gestation period of the eutherian viviparous reproductive system. However, postnatal muscle PEG11/RTL1 expression, which affects postnatal locomotor performance, may be species-specific because, in some species, pups require extensive maternal care for some time, while in others, pups can move quickly and spontaneously soon after birth.

It should be noted that *PEG11/RTL1* and *DLK1* are the major genes that cause the sheep callipyge phenotype because *PEG11/RTL1* was first identified in the course of the sheep callipyge study [2]. Both *PEG11/RTL1* and *DLK1* are critically involved in muscle development, and studies in transgenic mice also support this conclusion [59,72]. In sheep, *PEG11/RTL1* expression is relatively high until the late fetal stage, declines from just before birth, and is barely expressed after birth. The callipyge mutation recapitulates the normal fetal-like *PEG11/RTL1* expression program during postnatal development, and this may contribute to the emergence of the muscle hypertrophy phenotype [58].

### 2.3. PEG10 and PEG11/RTL1 in Neurological Disorders

KOS14 and TS14 patients exhibit certain neurodevelopmental disorders, such as developmental delay and/or intellectual disability and feeding difficulties in the former [51,53], and feeding difficulties, motor delay, early onset of puberty and mild intellectual disability in the latter [52,54]. *DLK1* is critically involved in the early onset of puberty in TS14 [56,57], while *PEG11/RTL1* is responsible for the other neurodevelopmental phenotypes in these patients, as *Peg11/Rtl1* Mat- and Pat KO mice, which overexpress and lack PEG11/RTL1 expression, provide strong evidence for this conclusion [33].

As in the case with muscle, *Peg11/Rtl1* mRNA expression in the central nervous system is restricted during the fetal to neonatal period, but at lower levels, and is barely detectable in adults. The PEG11/RTL1 protein is detected in the descending tracts, commissural fibers including the hippocampal commissure and corpus callosum, as well as the limbic system, i.e., the hippocampal fimbria, fornix, and medial amygdala nucleus (Figure 4) [33]. The corticospinal tract, one of the descending tracts, and the hippocampal commissure are mammalian-specific brain structures, whereas the corpus callosum is a eutherian-specific brain structure [73,74,75]. The corticospinal tract runs from layer V of the neocortex to the brainstem and spinal cord and is responsible for fine voluntary skilled muscle movements of the limbs, while the hippocampal commissure is involved in hippocampal-dependent memory output [33,73,76]. The corpus callosum is responsible for communication between the two hemispheres enabling faster transmission and integration of information from both sides [73,74,75]. These results suggest that *PEG11/RTL1* is deeply involved in the functional evolution of the eutherian brain. A high expression level of PEG11/RTL1 is also reported in the locus coeruleus (LC), while decreased neuronal excitability and increased delay of action potential onset and inward currents in LC neurons have been reported in *Peg11*/*Rtl1*-deficient mice [77].

*Peg11/Rtl1* Mat- and Pat-KO mice exhibit neurodevelopmental abnormalities corresponding to these expression sites, such as decreased spontaneous movement, increased anxiety-like behavior, and learning and memory impairments (Table 2) [33]. These symptoms suggest impairment of the corticospinal tract involved in trunk and limb movement, and/or the corpus callosum, hippocampal commissure, and medial amygdala nucleus. It is likely that they are also associated with the developmental delay and intellectual disability observed in KOS14 and TS14 patients [33]. It should be noted that maternally expressed *antiPeg11/antiRtl1* is also an important factor because it encodes seven miRNAs that regulate the *Peg11/Rtl1* mRNA levels via an RNAi mechanism [68,69,78]. 

Recently, *PEG10* was implicated in a certain neurological disorder, Angelman syndrome (AS) [35]. Pandya et al. demonstrated that the PEG10 protein accumulates in the neurons induced from AS patient iPS cells. AS is a severe neurodevelopmental genomic imprinting disorder, which is characterized by developmental delay, intellectual disability, severe language impairment, ataxia, and other symptoms [79,80], and caused by the paternal uniparental disomy of chromosome 15 and/or the mutations of maternally expressed *ubiquitin–protein ligase E3A* (*UBE3A*). They identified PEG10 as a target of UBE3A and suggested that *PEG10* may be critically involved in the pathophysiology of AS, although further work will be required to determine how PEG10 is mechanistically involved in AS.

*PEG10* has also been implicated in another neurological disorder, amyotrophic lateral sclerosis (ALS), a fatal neurodegenerative disease characterized by progressive loss of motor function, typically in middle age [81]. *UBQLN2*, a member of the ubiquilin family involved in proteasomal degradation, is the gene responsible for familial ALS [82]. Whiteley and colleagues demonstrated that UBQLN2 facilitates the proteasome-dependent degradation of the PEG10-ORF1/2 protein, and that PEG10-ORF1/2 is specifically upregulated in the spinal cord of ALS patients compared to healthy controls [36,37]. In addition, changes in gene expression in axon remodeling are induced by a nuclear-localized PEG10 fragment excised by its own self-cleavage activity in POL-like PEG10-ORF2. These results suggest that PEG10-ORF1/2 accumulation is an important contributor to ALS disease progression. Intriguingly, UBQLN2 is a marker protein of stress granules, where PEG10 localizes under the stress conditions observed in the AS study [35]. Therefore, it is possible that the pathogenic mechanism of AS and ALS may partially overlap and that PEG10 accumulation may also be a common mechanism causing other neuronal defects, including neurological disorders in which PEG10 has not yet been formally shown to be involved.

## 3. Retrovirus-Derived RTL/SIRH Genes as Eutherian-Specific Genes

The demonstration of essential roles for *PEG10* and *PEG11/RTL1* stimulated the screening of similar retrovirus-derived genes, and as a result, in eutherians and marsupials, 9 and 1 RTL/SIRH genes were, respectively, identified [5,20,21,83] (Figure 1). Except for therian-specific *PEG10*, all of the other SIRH/RTL genes, including *PEG11/RTL1,* are eutherian-specific genes. It is conceivable that all of the SIRH/RTL genes originated from the same retrovirus, because the encoded proteins exhibit 20~30% homology to the sushi-ichi retrotransposon GAG. It is also possible that eutherian-specific SIRH/RTL arose from cDNA retrotransposition from the PEG10 ORF1 (and sometimes ORF1/2) transcript because it is the oldest among the RTL/SIRH genes [62]. Alternatively, some could have arisen from *PEG11/RTL1,* because it might have emerged in a common therian ancestor like PEG10 but was lost in the marsupial lineage [63]. 

However, as shown in Figure 1, each protein has a unique amino acid sequence and length, and the SIRH/RTL genes have diverse functions. To date, 10 out of the 11 RTL/SIRH genes have been found to have essential and/or important functions in the current eutherian developmental system (Table 1).

### 3.1. RTL7/SIRH7 in the Placenta

In addition to maternal–fetal exchange and maternal tolerance of feto-paternal antigens, the placenta is a major endocrine organ during pregnancy [84]. *RTL7/SIRH7* (formal name: *Leucine Zipper Down-Regulated in Cancer 1* (*LDOC1*)) is another essential placental gene, like *PEG10* and *PEG11/RTL1.* It is expressed in trophoblast lineages in early placental development and regulates trophoblast differentiation (Figure 5A–J), and, thus, is deeply involved in several types of hormone production in trophoblast cells during pregnancy [85,86]. *Rtl7/Sirh7* KO placentas have an irregular boundary between the spongiotrophoblast and labyrinth layers, as well as a decreased number of SpTs, although the fetuses appear normal (Figure 5K). Female *Rtl7/Sirh7* KO exhibit delayed parturition due to residual progesterone (P4) in the serum on 18.5 dpc, one day before the parturition, and pups die due to inadequate maternal care [26]. Since all pups are viable when foster mothers are used, it is indicated that the problems are due to the KO mothers. 

As *Rtl7/Sirh7* is an X-linked gene, maternal transmission of the *Rtl7/Sirh7* KO allele results in a null phenotype in the placenta due to imprinted X-inactivation in the mouse placenta [87,88]. In *Rtl7/Sirh7* KO conceptuses, their placentas overproduce placental P4 (Figure 5I–K), leading to a delayed transition from placental lactogen (PL)1 to PL2 in the giant trophoblast cells in the KO placenta, which presumably in turn leads to delayed downregulation of maternal ovarian P4 production in the late pregnancy, resulting in delayed parturition.

Although the ovary was thought to be the major P4 producing organ in rodents throughout gestation [84], P4 production is observed in rodent placenta during 9.5 to 11.5 dpc when a temporal reduction in serum P4 level occurs due to a shift from the corpus luteum of pseudopregnancy to pregnancy (Figure 5L–N) [89,90], strongly indicating that placental P4 plays an important role in the maintenance of gestation at this critical stage and that *Rtl7/Sirh7* plays an important role in placental P4 production [26]. In eutherians, the P4 production in the corpus luteum is regulated by the pituitary and/or placenta, whereas it is autonomous in marsupials and monotremes [91]. Therefore, it will be of interest to elucidate how *Rtl7/Sirh7* functions in these processes at the molecular level because *Rtl7/Sirh7* is a eutherian-specific gene.

### 3.2. RTL4/SIRH11 in the Brain

*RTL4/SIRH11* (aka Zinc Finger CCHC Domain-Containing Protein 16 (ZCCHC16) is a causative gene in autism spectrum disorders (ASD) [92]. Lim et al. performed a comprehensive screening of patients with ASD and identified a family with a rare nonsense ZCCHC16 mutation leading to ASD in a male proband and his male sibling, as it is an X-linked gene [92]. 

In mice, *Sirh11/Zcchc16* KO mice exhibit increased impulsivity and decreased spatial memory, presumably due to low recovery of noradrenaline in the frontal cortex although they do not exhibit lethality or growth abnormalities [27]. They do not adapt to routine processes and tend to exhibit extreme behavior when transferred to a new environment. They display agitated movements in their cages when staff personnel enter the breeding room, and sometimes jump out when the cages are changed, even after a long breeding period. In particular, they remained hyperactive for 5 consecutive nights in the home cage activity test, while normal control mice gradually settled to lower levels of activity (Figure 6A). In the light/dark transition test, the latency before entering into the light chamber was significantly decreased, while the number of transitions was significantly increased, suggesting a reduced attention and/or enhanced impulsivity (Figure 6B). In the Y-maze test, KO mice exhibited a lower success rate, suggesting that they have a poor working memory (Figure 6C). This is likely due to low noradrenaline (NA) recovery in the frontal cortex (Figure 6D), because the locus coeruleus (LC) NA neurons have been reported to play important roles in attention, behavioral flexibility, and modulation of cognition [93,94,95] and their activation occurs in concert with the cognitive shifts that facilitate dynamic reorganization of target neural networks, allowing rapid behavioral adaptation to the demands of changing environmental demands [96], indicating that all of the behavioral defects of the *Sirh11/Zcchc16* KO mice are somehow related to a dysregulation of the noradrenergic system in the brain [27]. 

*RTL4/SIRH11* is a very important gene in neurodevelopment, and it is likely that it confers a critically important advantage both in the competition that occurs in daily life and in the evolution of the eutherian brain. However, because *RTL4/SIRH11* expression is very low in the brain in both humans and mice, it remains unclear exactly where the RTL4/SIRH11 protein is expressed and what its function is.

### 3.3. RTL8A, 8B, 8C/SIRH5, 6, 4 in the Brain

RTL8A, 8B, 8C/SIRH5, 6, 4 are triplet genes that encode almost identical proteins of 112 to 113 amino acids (aa). Their number (2–4, mostly 3, excluding pseudogenes) and aa sequence are well conserved in eutherians, suggesting they confer an evolutionary advantage. Therefore, it is of interest to know why they exist as multiple genes and what their function is. However, in most cases the RTL8A-C/SIRH5, 6, 4 genes within the same species exhibit higher homology to each other than other species, suggesting that they are not in a precise orthologous relationship in eutherians, presumably due to independent gene conversion events in each species. 

It was recently reported that the RTL8A, 8B, 8C/SIRH5, 6, 4 proteins accumulated together with the PEG10 protein in the neuronal cells differentiated from iPS cells of AS patients (see also Section 2.3.) [35]. AS is a neurodevelopmental genomic imprinting disorder which is characterized by delayed development, intellectual disability, severe speech impairment, ataxia, and other symptoms [79,80]. It is caused by paternal uniparental disomy of chromosome 15 and/or mutations of a maternally expressed UBE3A gene. This implies that the RTL8A, B, C/SIRH5, 6, 4, and PEG10 proteins are directly targeted by UBE3A in neuronal cells.

Rtl8a, b/Sirh5, 6 double KO mice exhibit late onset obesity and depression-like behavior, demonstrating they are also important brain genes [30]. These phenotypes correlate well with their protein expression sites, such as the hypothalamus and prefrontal cortex, the control centers for appetite [97,98,99] and depression [100,101,102], respectively. It is clear that decreased expression of RTL8A, B, C/SIRH5, 6, 4 proteins causes neuro-developmental disorders, and it is likely that overexpression of these proteins also plays some role in AS. Since no such behavioral abnormalities are observed in Rtl8c/Sirh4 single KO mice, it is likely that they have a gene dosage effect on neuronal development. Ongoing analysis of the Rtl8a, b, c/Sirh5, 6, 4 triple KO mice should provide an answer to this question.

### 3.4. RTL6/SIRH3, RTL5/SIRH8 and RTL9/SIRH10 Are Microglial Genes in the Brain

*RTL6/SIRH3* (aka LDOCKL), which encodes an extremely basic protein (pI = 11.15), is the most conserved gene among the RTL/SIRH genes, with a non-synonymous/synonymous (dn/ds) rate of less than 0.1. Despite its evolutionary importance, it has been very difficult to identify the RTL6/SIRH3 protein because of the lack of effective antibodies. Analysis of Rtl6-CV KI mice, in which a Venus ORF is integrated into the endogenous Rtl6/Sirh3 locus immediately after the C-terminus, demonstrated that the RTL6/SIRH3 protein is expressed in the central nervous system during development and that it is secreted by microglia and responds to LPS (Figure 7A). It was subsequently demonstrated in Rtl6/Sirh3 KO mice that the RTL6/SIRH3 protein functionally protects against bacteria by removing lipopolysaccharide (LPS) [29]. Thus, the Rtl6/Sirh3 gene plays an important role in the innate immune system in the brain. Because LPS is a highly acidic substance and an extremely dangerous pathogen, this explains why RTL6/SIRH3 is extremely basic and highly conserved in eutherians (Figure 7B), because the role of RTL6/SIRH3 in LPS removal is critical.

*RTL5/SIRH8* (aka *Retrotransposon Gag Domain-Containing Protein* (*RGAG4*)) is phylogenetically related to *RTL6/SIRH3* and well conserved in eutherians, despite there being some exceptions. It encodes a larger protein which covers the entire RTL6/SIRH3 and is strongly acidic (pI = 4.39), and functions as another microglial gene in the innate immune system against viruses by removing double-stranded RNA from the brain [29]. Analysis of *Rtl5-CmC* KI mice in which an mCherry ORF is integrated into the endogenous *Rtl5/Sirh8* locus immediately after the C-terminus demonstrated that the RTL5/SIRH8 protein is also expressed in the brain and that the RTL5/SIRH8 protein is likewise secreted by microglia and responds to double-stranded (ds) RNA (Figure 7C). Subsequently, it was demonstrated in *Rtl5/Sirh8 KO* mice that the RTL5/SIRH8 protein functions against viruses by removing dsRNA [29]. This explains why *RTL5/SIRH8* is strongly acidic and well conserved in eutherians (Figure 7D), because the role of *RTL5/SIRH8* in dsRNA removal is also of critical importance.

Using the same approach of combining Venus KI mice and KO mice, *RTL9/SIRH10* (aka *RGAG1*) was demonstrated to be another microglial gene that is actively protective against fungi by reacting to zymosan, the cell wall of fungi [31]. It encodes a large protein comprising two herpes virus-derived domains in addition to the GAG-like domain. The role of the first two regions remains unknown, but the latter is essential for zymosan removal. Unlike RTL6/SIRH3 and RTL5/SIRH8, RTL9/SIRH10 is restricted to the lysosomes of microglia, where the zymosan is ultimately taken up and degraded. Thus, at least three RLT/SIRH genes are involved in the clearance of bacterial, viral, and fungal pathogens from the brain, suggesting that these genes must have critically contributed to the evolution of the innate immune system in eutherians [29,31]. 

### 3.5. Relationship between RTL/SIRH Genes and Extraembryonic Tissues

All of these studies clearly demonstrate that domestication of retrovirus-derived genes made important contributions to the generation of eutherian-specific features in the placenta and brain. Microglia originate from the yolk sac during early development and eventually become permanently resident in the brain. Therefore, it is of interest to notice that the placenta and yolk sac, the extraembryonic tissues in intrauterine development, evidently serve as origin sites for the incubation of such retrovirus-derived genes, including both *PEG10* and *PEG11/RTL1*, in the course of mammalian evolution (Figure 8) [22,29,31,46]. It is known that the extraembryonic tissues have lower levels of DNA methylation than the embryos; therefore, endogenous retroviruses and retrotransposons can express there even at a low level. This may have promoted the domestication (exaptation) of retrovirus-derived genes in eutherians.

## 4. Two Types of Exapted Genes: Acquired RTL/SIRH Genes and Captured *Syncytin* Genes

As mentioned in the introductory section, *PEG10*, *PEG11/RTL1*, and other RTL/SIRH genes are thought to be derived from GAG and POL of an extinct retrovirus. *PEG10* emerged between 164 and 148 MYA, after the diversification from monotremes and before the split between eutherians and marsupials [62], while *PEG11/RTL1* and the other RTL/SIRH genes emerged between 148 and 120 MYA, after the diversification from marsupials and before the emergence of common eutherian ancestor [22,63]. They completely lost LTR sequences at both ends and also an ENV gene, which were present at the time of the original retroviral insertions. Their encoded proteins share 20 to 30% homology to the sushi-ichi retrotransposon but are completely different proteins from the original GAG and POL, each with its own unique aa sequence and novel function. Hence, they are referred to as genes acquired from a retrovirus [22]. 

In contrast, *ENV*-derived *syncytins* were sequentially domesticated in a lineage-specific manner after the establishment of the eutherians. For example, *syncytin-1* and *syncytin-2* emerged 40 and 20 MYA in primates from different retroviruses, respectively [3,4,9]. They retain almost all of the ENV sequences of the original retrovirus and, therefore, have fusion activity, which is important for syncytiotrophoblast cell fusion in the placenta. The LTRs, GAG, and POL sequences often persist in a remnant form due to severe mutations. Therefore, it is hypothesized that the *syncytin* gene was replaced several times in each eutherian lineage by a newly integrated *ENV* gene having superior fusion activity [103,104]. Therefore, they are called captured genes [103]. Thus, there are two distinct ways to domesticate retrovirus-derived genes.

## 5. Conclusions and Future Prospects

The RTL/SIRH genes contributed to evolutionary modification, such as the emergence of the placenta, the development of highly organized brain functions and the improvement of the brain’s innate immune system in therian- and/or eutherian-specific ways. However, this may be just the tip of the iceberg, and more diverse functions will be elucidated in the future in other as-yet-to-be-identified retrovirus-derived genes. The concept of retrovirus-derived genes will also be important in determining how many protein-coding genes exist in humans. It is likely that there are many more primate- or human-specific genes derived from retroviruses and that they may play an important role in certain primate- and/or human-specific traits.

The exaptation of retrovirus-derived genes implies the robustness and/or plasticity of the organism. The fact that many retrovirus-derived genes have been identified in eutherians suggests that eutherians may have greater robustness and/or plasticity than other vertebrates, or it may simply reflect the fact that the eutherian genome has been more extensively analyzed than the others. It is likely that extraembryonic tissues, such as the placenta and yolk sac, served as origin site for the retrovirus-derived genes because they have lower levels of DNA methylation, which represses the expression of endogenous retroviruses and retrotransposons. This may have increased the chances for exaptation of retrovirus-derived genes in eutherians. 

It is of considerable interest to establish how the retrovirus-derived genes function and why the GAG proteins are selected for such diverse roles. What is the function of their CCHC RNA-binding motif and DSG protease motif? Do they function as enzymes or as structural proteins, or as virus-like particles (VLPs)? Recently, PEG10 and other GAG-like proteins have been shown to self-assemble into VLPs [105,106]. Taking advantage of this property, Segel et al. developed a novel specific RNA deliver system called selective endogenous encapsidation for cellular delivery (SEND) [106]. However, caution should be exercised in its use because overexpression of PEG10 itself exerts untoward effects, as seen in AS and ALS. The history of how they became mutated and were selected is also of great interest, as is the question of how they made an evolutionary contribution. In conclusion, the genes derived from the retroviruses will open a new window on the relationship between development and evolution in organisms, and the question of how ontogeny and phylogeny are interrelated.

## Figures and Tables

**Figure 1 biomolecules-13-01436-f001:**
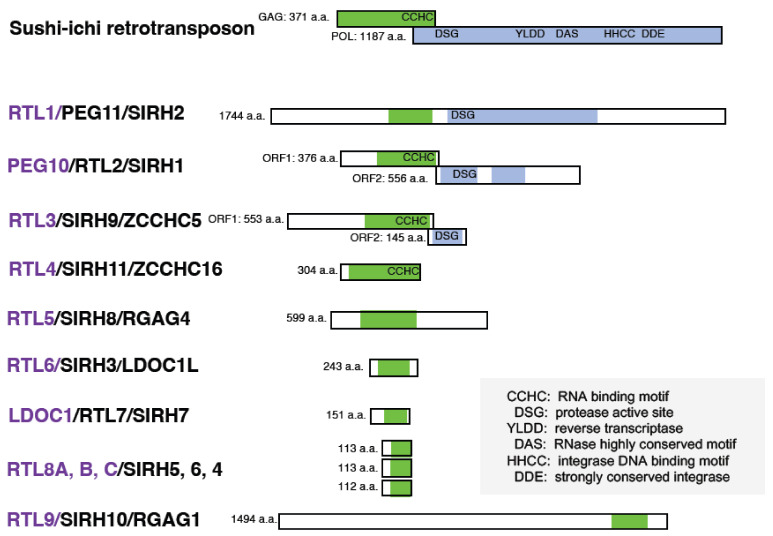
RTL/SIRH proteins in mice. There are 11 retrotransposon Gag-like/sushi-ichi retrotransposon homologs (RTL/SIRH) genes in eutherians that encode proteins with homology to the sushi-ichi retrotransposon GAG (green) and POL (light blue). The mouse RTL/SIRH proteins are shown as representative examples. The CCHC RNA binding motif and/or the DSG viral protease motif are conserved in certain RTL/SIRH genes. Purple: formal name, black: aliases. Sushi-ich was originally isolated from *Takifugu rubripes* [23,24] and belongs to the vertebrate-like *Ty3/Gypsy,* the V-clade in the chromovirus lineage [25].

**Figure 2 biomolecules-13-01436-f002:**
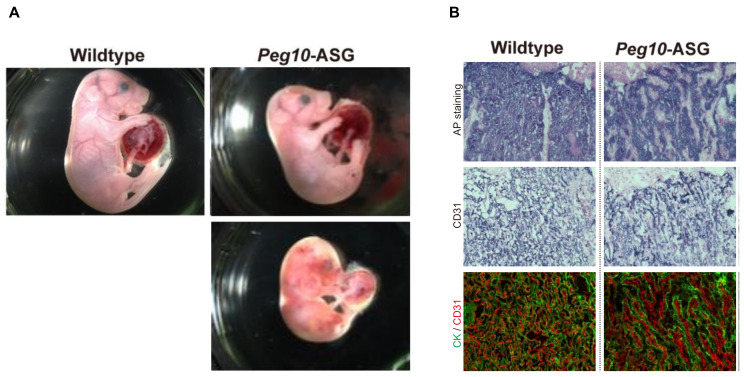
Phenotypes of *Peg10*-ASG embryos and placentas. Half of the *Peg10*-ASG embryos exhibit lethality at 18.5 dpc (**A**) with severe damage to the placental vasculature (**B**). Alkaline phosphatase (AP) is positive in the majority of chorionic trophoblast cells. CD31 (PECAM) is an endothelial cell marker and cytokeratin (CK) is a pan-trophoblast marker. Reproduced with permission from [34]. Copyright 2021 The Company of Biologist Ltd.

**Figure 3 biomolecules-13-01436-f003:**
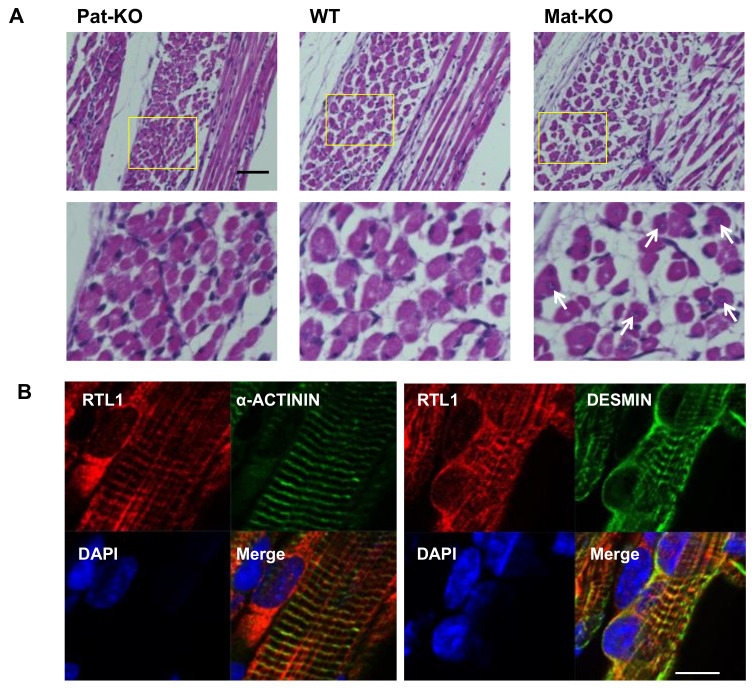
PEG11/RTL1 protein expression in muscle. (**A**) HE staining in neonatal intercostal muscle after super fix treatment (upper column) and higher magnification views of the yellow boxes in the upper column (lower column): Pat-KO (left), wild type (middle), and Mat-KO (right). The white arrows in the Mat-KO column indicate muscle fibers with centrally located nuclei. Scale bars: 50 μm. (**B**) Immunofluorescence staining of Mat-KO neonatal forelimb muscle using anti-PEG11/RTL1 (red), anti-α-ACTININ (green, left panels), and anti-DESMIN (green, right panels) antibodies. PEG11/RTL1 is expressed in a typical striated pattern in the striated muscle, with a pattern strikingly similar to that of α-ACTININ located at the sarcomeric Z-disc (left panels); however, they are not merged. In contrast, PEG11/RTL1 is partially merged with DESMIN, one of a sarcomeric cytoskeleton showing some of the connections between membranes and sarcomeres at the Z-disc. Reproduced with permission from [29]. Copyright 2020 The Company of Biologists Ltd.).

**Figure 4 biomolecules-13-01436-f004:**
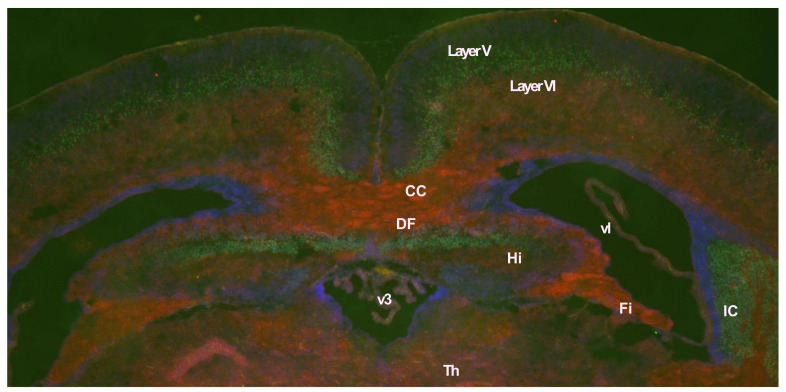
PEG11/RTL1 protein expression in brain. Immunofluorescence staining of a coronal section of the neonatal brain using anti-PEG11/RTL1 (red) and anti-CTIP2 (green) antibodies. Blue: DAPI. CTIP2 is a marker of the cerebral cortex layer V. CC: corpus callosum, DF: dorsal fornix, Fi: fimbria, Hi: hippocampus, IC: internal capsule, layer V: the fifth layer of cerebral cortex, layer VI: the sixth layer of cerebral cortex, Th: thalamus, v3: the third ventricle, vl; lateral ventricle. Reproduced with permission from [33]. Copryright 2021 Molecular Biology Society of Japan and John Wiley & Sons Australia, Ltd.

**Figure 5 biomolecules-13-01436-f005:**
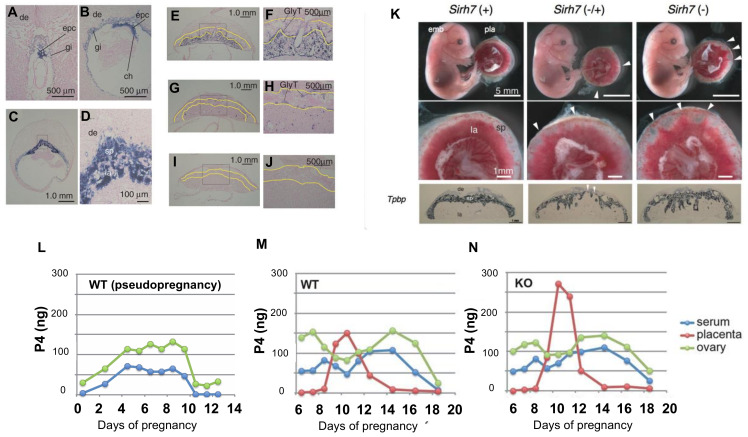
Placental expression of *Rtl7/Sirh7* mRNA and placental P4 production. **A**–**J**: In situ hybridization of *Rtl7/Sirh7* in the placentas at 7.5 (**A**), 8.5 (**B**), 9.5 (**C**,**D**), 12.5 (**E**,**F**), 15.5 (**G**,**H**), and 18.5 (**I**,**J**) dpc in wild type mice. The boxed areas in (**C**,**E**,**G**,**I**) are magnified in (**D**,**F**,**H**,**K**), respectively. The yellow lines enclose the area of the spongiotrophoblast layer. ch, chorion; epc, ectoplacental cone; gi, trophoblast giant cells (TGCs); sp, spongiotrophoblast layer; la, labyrinth layer; de, decidua; GlyT, glycogen trophoblast cells. (**K**) *Rtl7/Sirh7* KO fetuses and placentas at 15.5 dpc. The white arrowheads indicate the regions where the labyrinth layers almost reached the maternal decidua. Despite the placental abnormalities, the *Rtl7/Sirh7* KO fetuses appeared normal. L–N: P4 levels in ovary (green), placenta (red), and serum (blue) of pseudo-pregnant wild type (**L**), pregnant wild type (**M**), and pregnant *Rtl7/Sirh7* KO mice (**N**). Reproduced with permission from [26]. Copyright 2014 The Company of Biologists Ltd.

**Figure 6 biomolecules-13-01436-f006:**
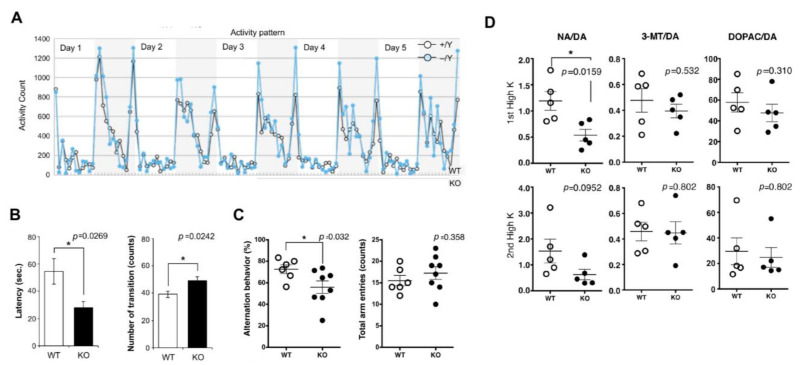
Abnormal behaviors of *Rtl4/Sirh11* KO mice. (**A**) Daily activity test. Activity counts of WT and *Rtl4/Sirh11* KO mice are shown as black and blue lines, respectively. (**B**) Light/dark transition test. The latency to enter into the light chamber was significantly decreased (left), while the number of transitions was significantly increased compared to the wild type (right). * indicates *p* < 0.05 (**C**) Y-maze test. KO mice exhibited a lower level of alternation (left), although the total number of arm entry events was the same (right). (**D**) Microdialysis analysis in the prefrontal cortex in the cerebrum. The levels of DA, NA, 3-MT, and DOPAC in the prefrontal cortex were measured after perfusion of high potassium-containing artificial cerebrospinal fluid. The ratio of the DA metabolites, namely NA, 3-MT, and DOPAC, to DA is shown. DA: dopamine, NA: noradrenaline, 3-MT: 3-methoxytyramine, DOPAC: 3, 4-dihydroxyphenylacetic acid. Reproduced with permission from [27]. Copyright 2015 PLOS.

**Figure 7 biomolecules-13-01436-f007:**
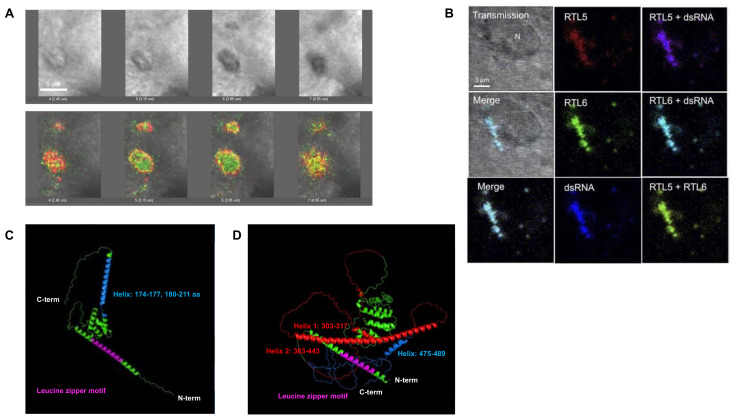
RTL6/SIRH3 and RTL5/SIRH8 proteins function in pathogen elimination. (**A**) Large complexes were formed by the interaction between the RTL6 protein and LPS. Two sets of a sequence of photographs, transmission (top) and fluorescence, are shown at 0.7 μm intervals. Note that this complex also includes RTL5/SIRH8. (**B**) RTL6/SIRH3 and RTL5/SIRH8 proteins formed a large complex with dsRNA near the nucleus of a microglial cell. (**C**,**D**) AlphaFold2 prediction of the 3D structure of RTL6/SIRH3 (**C**) and RTL5/SIRH8 (**D**) proteins. Green indicates amino acids without specfic function(s) in (**C**,**D**) Both proteins have a leucine-zipper motif (pink) at the N-terminus. The RTL6/SIRH3 protein has an extremely basic helix at the C-terminus, while the acidic and basic domains are spatially separated in the RTL5/SIRH8 protein. Reproduced with permission from [29]. Copyright 2022 The Company of Biologists Ltd.

**Figure 8 biomolecules-13-01436-f008:**
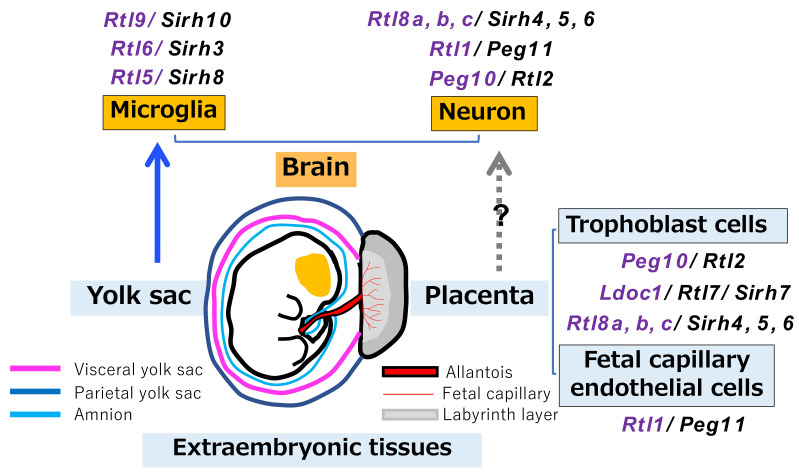
Extraembryonic tissues may be the birthplace of RTL/SIRH genes. To date, 9 out of 11 RTL/SIRH genes have been shown to be domesticated in extraembryonic tissues, such as the placenta and yolk. *Rtl4/Sirh11* is expressed in the brain, but it remains unknown which cells express it. It also remains unknown how several placental RTL/SIRH genes gained neuronal expression in the brain. See also Table 1. Reproduced with permission from [29]. Copyright 2022 The Company of Biologists Ltd. Information of *Rtl9/Sirh10* and *Rtl8a, b, c/Sirh4, 5, 6* are added.

**Table 1 biomolecules-13-01436-t001:** Expression sites of RTL/SIRH proteins, possible relationship to human disease. and abnormal phenotypes in KO mice. ND: not detected, NK: function not known, −: no data, *: possible. No information on the RTL3/SIRH9 protein expression. The RTL4/SIRH11 protein is expressed in the brain, but it remains unclear which cells express it. See text for a more detailed description of phenotypes. ALS: amyotrophic lateral sclerosis, AS: Angelman syndrome, ASD: autism spectrum disorders, ASG: point mutation of the DSG protease active site, dsRNA: double-stranded RNA, FC: frontal cortex, KOS14: Kagami–Ogata syndrome, LPS: lipopolysaccharide, Mat-KO: maternally derived KO, NA: noradrenaline, Pat-KO: paternally derived KO, PWS: Prader–Willi syndrome, TS14: Temple syndrome.

	Formal Gene Name	Aliases		Placenta	Brain		Muscle	Human Diseases	Behavior Abnormalities	Rtl/Sirh KO Mouse	Phenotypes
					Neuron	Microglia					
**1**	*RTL1*	*PEG11*	*SIRH2*	◯	◯		◯	KOS14, TS14		*Rtl1/Peg11* Pat-KO	Late fetal/neonatal lethality, TS14-like phenotypes in placenta, muscle and brain
										*Rtl1/Peg11* Mat-KO	Neonatal lethality, KOS14-like phenotypes in placenta, muscle and brain
**2**	*PEG10*	*RTL2*	*SIRH1*	◯	◯		NK	AS *, ALS *		*Peg10* KO	Early embryonic lethality due to severe placental defects
										*Peg10* ASG-mutant	Late fetal/neonatal lethality due to placental defects
**3**	*RTL3*	*ZCCHC5*	*SIRH9*	–	–		–		–	*Rtl3/Sirh9* KO	No data
**4**	*RTL4*	*ZCCHC16*	*SIRH11*	ND	◯		ND	ASD	◯	*Rtl4/Sirh11* KO	Increased impulsivity, reduced attention and spacial memory possibly due to low recovery rate of NA in FC
**5**	*RTL5*	*RGAG4*	*SIRH8*	NK		◯	NK		◯ *	*Rtl5/Sirh8* KO	Defects in innate immunity against viruses (dsRNA)
**6**	*RTL6*	*LDOC1L*	*SIRH3*	NK		◯	NK		◯ *	*Rtl6/Sirh3* KO	Defects in innate immunity against bacteria (LPS)
**7**	*LDOC1*	*RTL7*	*SIRH7*	◯	◯ *		NK		◯	*Rtl7/Sirh7* KO	Delayed parturition due to abnormal placental endocrine function, poor maternal care
**8**	*RTL8A, B, C*	*CXXA, B, C*	*SIRH5, 6, 4*	◯	◯ *		NK	AS *, PWS *	◯	*Rtl8a, b/Sirh5, 6* DKO	Increased depression-like behavior and anxiety, late onset obesity, poor maternal care
9	*RTL9*	*RGAG1*	*SIRH10*	NK		◯	NK		◯ *	*Rtl9/Sirh10* KO	Defects in innate immunity against fungi (Zymosan)

**Table 2 biomolecules-13-01436-t002:** Abnormal behaviors observed in *Peg11* Pat- and Mat-KO mice.

Behavioral Tests	WT vs Pat-KO	WT vs Mat-KO	Relationship between Behavioral Analysis and PEG11/RTL1 Expression
Open Field Test	Decreased spontaneous movement		Impaired functions of the CC and cortical layer V possibly due to reduced information exchange
Elevated Plus Maze Test	Increased anxiety-like behavior	NS	Increased anxiety due to impaired cortex, amygdala and hippocampus
Light/Dark Transition Test	Mild increament of anxiety-like behavior		Increased anxiety due to impaired cortex, amygdala and hippocampus
Fear Conditioning Test: hippocampus-dependent memory	NS	Impaired hippocampus-dependent memory	Increased anxiety due to impaired hippocampus-dependent fear responses
Fear Conditioning Test: amygdala-dependent memory	Impaired amygdala-dependent memory		Increased anxiety due to impaired amygdala-dependent fear responses
Morris Water Maze Test	NS	Severely impaired spatial memory	Impaired hippocampus-dependent learning

*Peg11* Pat- and Mat-KO mice exhibit phenotypes similar to TS14 and KOS14 patients, respectively.

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
