# Peer review of "Retrovirus-Derived RTL/SIRH: Their Diverse Roles in the Current Eutherian Developmental System and Contribution to Eutherian Evolution"

_biomolecules, 2023, doi:10.3390/biom13101436_

Round 1

Reviewer 1 Report

The manuscript entitled "Retrovirus-Derived RTL/SIRH Genes: Their Diverse Roles in the Current Eutherian Developmental System and Contribution to Eutherian Evolution" is a review article summarizing a series of the authors’ previous studies clarifying the function of RTL/SIRH genes and discussing their roles in eutherian development and evolution. The manuscript is well structured, clear to read and introducing the intriguing and important works comprehensively. I suppose this review would appeal a broad range of readers to understand the importance of retrovirus-derived genes for the evolution of mammals. Here are a few minor points should be addressed to improve the manuscript.

1. In the line 49, “SIHR” should be “SIRH” and “former name” should be “formal name”. 

2. In the line 59, I suggest writing “other RTL/SIRH genes” as PEG10 and PEG11/RTL1 are also included in RTL/SIRH genes.

3. In the Table 1, it is difficult for me to understand how “- (no data)” and “?” are different. Some explanation or unification will be needed.

4. It is difficult to understand what the sentence starting from the line 200 is saying. According to this section, I understand that acquisition of PEG11/RTL1 contributed to stabilize and strengthen the muscle, but the sentence discusses the weak muscular strength and reduced movement of the fetus.

5. In the line 204, “PEG11 expression” should be “PEG11/RTL1 expression”.

6. In the line 205, some reason explaining why it may be species-specific should also be described.

7. In the line 245, “commissureare” should be “commissure are”.

8. In the line 376, “They” should be non-Italic.

9. In the line 389, “andtheir” should be “and their”.

10. In the Figure 7, Fig.7B should be Fig.7C and vice versa.

Author Response

Thank you very much for your critical reading and good evaluation to our manuscript.

We addressed all the comments as below:

  1. In the line 49, “SIHR” should be “SIRH” and “former name” should be “formal name”.

We amended them.

  1. In the line 59, I suggest writing “other RTL/SIRH genes” as PEG10 and PEG11/RTL1 are also included in RTL/SIRH genes.

We amended it.

  1. In the Table 1, it is difficult for me to understand how “- (no data)” and “?” are different. Some explanation or unification will be needed.

Thank you for your point. We distinguished ND (not detected), NK (not known) and - (no data).

  1. It is difficult to understand what the sentence starting from the line 200 is saying. According to this section, I understand that acquisition of PEG11/RTL1 contributed to stabilize and strengthen the muscle, but the sentence discusses the weak muscular strength and reduced movement of the fetus.
  2. In the line 204, “PEG11 expression” should be “PEG11/RTL1 expression”.
  3. In the line 205, some reason explaining why it may be species-specific should also be described.

Thank you again for your points. We agreed with the reviewer 1’s comments and changed the sentences as below:

We assume that the intrinsically weak muscular strength and reduced movement of the fetus is necessary for a safe pregnancy because it is beneficial to both the mother and fetus. Then, PEG11/RTL1 may be required for such regulation by interacting or interfering with DESMIN and is therefore a well conserved feature of eutherians, suggesting that PEG11/RTL1 expression in the fetal muscle is an adaptation to the long gestation period of the eutherian viviparous reproductive system. However, postnatal muscle PEG11/RTL1 expression, which affects postnatal locomotor performance, may be species-specific because in some species, pups require extensive maternal care for some time, while in others, pups can move quickly and spontaneously soon after birth.

  1. In the line 245, “commissureare” should be “commissure are”.
  2. In the line 376, “They” should be non-Italic.
  3. In the line 389, “andtheir” should be “and their”.

We amended them.

  1. In the Figure 7, Fig.7B should be Fig.7C and vice versa.

We amended them.

Reviewer 2 Report

This manuscript summarizes the function of retrovirus-derived RLT/SIRH genes in eutherian development. This review is well-organized and provides an interesting perspective regarding these ERV-derived genes. I generally agree with the publication of this manuscript. I only have one suggestion to the authors before publication. It will be clearer to the reader if the authors include a table listing: 1. RLT/SIRH genes are from which exact subtype of ERVs; 2. Knockout phenotype of each gene.

Author Response

Thank you very much for your favorable comments to our manuscript.

It will be clearer to the reader if the authors include a table listing: 1. RLT/SIRH genes are from which exact subtype of ERVs; 2. Knockout phenotype of each gene.

Thank you for your points.

  1. As all the RTL/SIRH genes exhibit homology to the suchi-ichi LTR retrotransposon. Therefore, we added its explanation in the Figure 1 legend as below:

Sushi-ich belongs to the vertebrate-like Ty3/Gypsy, the V-clade in the chromovirus lineage [105].

  1. Llorens, C., Muñoz-Pomer, A., Bernad, L.,Botella,H., Moyam, A.(2009). Network dynamics of eukaryotic LTR retroelements beyond phylogenetic trees. Biol Direct 4, 41. doi: 10.1186/1745-6150-4-41

2. According to the reviwer 2’s suggestion, we rearranged Table 1 to include the KO mouse phenotypes.